# Prediction Score for Identification of ESBL Producers in Urinary Infections Overestimates Risk in High-ESBL-Prevalence Setting

**DOI:** 10.3390/antibiotics14090938

**Published:** 2025-09-17

**Authors:** Jorge Alberto Cortés, Julián Antonio Niño-Godoy, Heidi Johanna Muñoz-Latorre

**Affiliations:** 1Department of Internal Medicine, School of Medicine, Universidad Nacional de Colombia, Sede Bogotá, Bogotá 111321, Colombia; 2Infectious Disease Service, Hospital Universitario Nacional, Sede Bogotá, Bogotá 111321, Colombia; 3Department of Infectious Diseases, Clínica Reina Sofía, Colsanitas, Bogotá 110911, Colombia

**Keywords:** urinary tract infection, beta-lactamases, validation study, *Escherichia coli*, Colombia

## Abstract

**Background/Objectives:** Urinary tract infections (UTIs) caused by extended-spectrum beta-lactamase (ESBL) Enterobacterales have become more frequent. Therefore, strategies for assessing the risks posed by ESBL-producing infections have been developed, creating the need for local validation. The aim of this study was to validate the scoring system designed by Tumbarello et al. to identify ESBL producers in patients with a UTI that require hospital care in a region with a high prevalence of ESBL *Escherichia coli*. **Methods**: A retrospective cohort study was conducted in a third-level hospital in Bogotá (Colombia) between 2013 and 2020.The study included 817 patients, who were hospitalized due to pyelonephritis and treated with cefuroxime (the first-line antibiotic according to local guidelines), with an ESBL frequency of 9.68%. Diagnostic performances were estimated for a modified version of Tumbarello’s score (omitting admission from another healthcare facility) evaluating the area under the curve (AUC) for ESBL presence with respect to resistance to second- and third-generation cephalosporins. **Results**: With an index cut-off of ≥6, the score showed a sensitivity of 18% and a specificity of 83%. The AUC for this cut-off was 0.47. This threshold index could not efficiently predict either third- (AUC = 0.49) or second-generation cephalosporin resistance (AUC = 0.51). **Conclusions**: In Colombia, a region with a high prevalence of ESBL *E. coli* producers, as the use of the Tumbarello index would result in excessive utilization of wide-spectrum antibiotics, it is not recommended in this specific scenario for UTIs. Further studies are required in order to develop accurate tools to assess the risk of ESBL producers in high-prevalence settings.

## 1. Introduction

Urinary tract infections (UTIs) are frequent community-acquired infections caused by different species of Enterobacterales. Topping the list of the most frequent causes of UTIs, *Escherichia coli* accounts for 75 to 95% of all cases. Depending on the series, it is followed by *Klebsiella pneumoniae*, with 3.5 to 13%, and *Proteus* spp., ranging from 3.5 to 6% [1]. These microorganisms have acquired various beta-lactam-antibiotic enzymes that can hydrolyze several beta-lactam antibiotics. One important group with the capacity to affect the activity of third-generation cephalosporins is named extended-spectrum beta-lactamases (ESBLs) [2], which limit the efficacy of other first-line antibiotic agents. The burden associated with this kind of antimicrobial resistance has been estimated to have an important impact on morbidity and mortality worldwide [3].

Resistance mechanisms can easily spread in both hospitals and communities. In Colombia, considered an endemic area, the prevalence of this resistant phenotype has steadily increased to concerning levels, rising from 10.1% in 2011 to 15% in 2019 [3,4] among isolates of *E. coli* identified in urinary samples [5]. This trend has been associated with horizontal transmission between bacteria through plasmids mainly from the CTX-M family [6,7,8].

Due to the potential failure of cephalosporins among ESBL infections, different prediction scores have been developed [9]. One of the first and most frequently used scores is the one developed by Tumbarello et al. [10]. The expected benefit of using such a score is that it may allow for the implementation of policy on the use of empirical broad-spectrum antibiotics for the treatment of infections that can be caused by these multidrug-resistant microorganisms, thereby improving patients’ outcomes.

The aim of this study is to validate the scoring system proposed by Tumbarello et al. for identifying ESBL producers in patients with community-acquired pyelonephritis that required hospital care in Colombia, a high-prevalence region with respect to ESBL-producing *E. coli*.

## 2. Results

We identified 817 patients with urinary tract infections caused by *E. coli* that required hospitalization. Of these cases, 113 (13.8%) were non-susceptible to second-generation cephalosporins, 110 (13.46%) were non-susceptible to third-generation cephalosporins, and 79 (9.6%) were identified as ESBL producers according to a phenotypic test.

We determined the score’s diagnostic performance in predicting ESBL non-susceptibility to second- and third-generation cephalosporins. The main patient demographics are described in Table 1.

Notably, the overall antibiotic usage was 18.9%, and no statistical difference was found in the frequency of antibiotic exposure between ESBL and non-ESBL infections.

The scoring system’s diagnostic performance is presented in Table 2. When using the suggested cut-off of 6 points, the sensitivity and specificity would be 18.9% and 83.1%, respectively, with an accuracy of 76.9%. The estimated ROC curve was 0.47 (95% CI 0.40–0.54). We also calculated the score’s performance in predicting resistance against third-generation cephalosporins. With a 6-point cut-off, the sensitivity and specificity would be 20.0% and 83.5%, respectively, with an accuracy of 74.9%.

Regarding the ROC-AUCs, the results for predicting ESBL infection, second-generation cephalosporin non-susceptibility, and third-generation cephalosporin non-susceptibility were 0.47 (95% CI 0.40–0.54, Figure 1), 0.50 (95% CI 0.45–0.57), and 0.49 (95% CI 0.43–0.55), respectively. With the sample provided, the power for this ROC-AUC was 0.23.

## 3. Discussion

In this retrospective cohort of 817 patients, we found that the scoring system for ESBL producers in hospitalized patients with pyelonephritis caused by an *E. coli* infection failed to appropriately discriminate the patients carrying ESBL-producing bacteria. This scoring system, which is based on healthcare and antibiotic exposure, yields high proportions of false negatives and false positives, limiting its use in common clinical practice in countries like Colombia, wherein such resistance is highly endemic.

Interestingly, differences between the group of patients harboring ESBL-producing isolates and those who did not were not many. The frequency of previous antibiotic use showed little difference between the groups (near 4%). Although this could be due to information bias due to the retrospective nature of the study, it also reflects the reality: patients do not recall with precision the previous use of antimicrobial agents, and information could be lacking because of missing medical records, previous hospitalization in other healthcare facilities, and other reasons. The other variable with a difference between the groups was the previous use of a urinary catheter, which might also be related to males, who more frequently require this kind of device for prostate conditions and procedures.

Antibiotic resistance via ESBL genes can be induced and transmitted in settings outside hospital environments. Multiple studies conducted in different countries have reported ESBL enterobacterales in household and food-production animals and in various environmental sources [11]. Other studies have found ESBL producers in poultry, processed meats sold for human consumption, biofilms, and healthy household animals [8,11,12,13]. These environmental sources are predominantly transmitted through plasmids, transposons, and integrons, which have been found to be similar to those in human feces, suggesting that they are a dominant facilitator of the spread of ESBLs in the human microbiome [8,12,14].

A study evaluating the risk of acquiring ESBL infections when using standard precautions vs. contact isolation conducted under similar conditions regarding compliance with hand hygiene, glove and gown use, and antimicrobial consumption showed that contact isolation had no significant beneficial effects on controlling the spread of ESBLs, implying that transmission is common both outside and inside the hospital [15].

Unfortunately, our results were negative. The performance of the score with a ≥6 cut-off might result in an incorrect decision; however, the overall balance between sensitivity and specificity is poor, and no right cut-off point can be recommended. Moreover, the ROC curve shows that the prediction is not better than chance alone. There is a need for a method to accurately predict whether patients harbor resistant microorganisms [16]. Septic and cancer patients with severe infections rely on an appropriate antibiotic for infection control and survival [17]. High levels of resistance are related to poorer outcomes in Latin American countries [18]. In the context of urinary tract infections, where the risk of mortality is lower than that for other common infections, the risk of using broad-spectrum antibiotics because of a false perception of resistance risk might contribute to further antimicrobial resistance and poorer outcomes. Recent publications have shown that there is no universal benefit of the administration of broader-spectrum antibiotics to patients with severe infections [19]. Treating urinary tract infections with non-susceptible antibiotics might increase the risk of late recurrence but not clinical failure [20,21]. Predicting a low risk of resistance might promote the use of lower-spectrum antibiotics. In the meantime, the use of antibiotics of a lower spectrum in the majority of cases seems to be the right approach. The use of antibiotics with a high concentration in urine, such as cefazoline, cefuroxime, or ceftriaxone [22], has a low risk of failure [20,21,23] and lower risk of affecting the resistance microenvironment of the patient, especially when used for a short period of time [24].

The need to predict a resistance pattern has been of interest in infectious diseases. The score proposed by Tumbarello et al. [10] was one of the first to be developed. Since then, other strategies have been used in different environments. Difficulties arise from balancing the simplicity of the information required for the prediction and the need for accurate results. Other variables might contribute to resistance and could be important in the prediction. A Spanish study conducted in a nursing home identified chronic renal disease, neoplasia, non-urological procedures, and recurrent urinary tract infections as risk factors for resistance [25]. A risk stratification model was developed using a huge data base for non-complicated urinary tract infection [26]. With moderate accuracy, it categorizes patients in risk categories and allows understanding of the effect of specific antibiotic use on resistance patterns. Recently, models using artificial intelligence (AI) have surpassed the traditional models. A small model using decision trees and AI had a very good accuracy (0.93) for predicting ESBL [27]. A complex model with 39 variables was established and used in an emergency department for predicting resistance for ciprofloxacin [28]. With a better performance, it also showed that the number of variables required to make a better prediction is between 15 and 20. It resulted in lesser use of inappropriate therapy. Taking these studies together, it seems that an AI system with a higher number of variables is required to make a better prediction. Unfortunately, this calls for a less simple decision-making process. On the other hand, geographic information might add vital data for resistance in the community. A previous study in Colombia showed that community transmission of ESBL isolates might have a geographical pattern [29]. Together with food data (preferences, procedence), such information might result in a better prediction of resistance in real community-acquired infections [12].

One of the limitations of our study pertains to the quality of the clinical records employed, which could have been incomplete, leading to underestimation of the score points for some patients. Patients tend to forget about or assign low priority to antibiotic consumption. We selected only patients with *E. coli* isolates as this microorganism is the main cause of urinary tract infections, as well as one of the most frequent producers of this kind of enzyme [1,30]. However, it might not represent all cases of urinary tract infections, which can be caused by other microorganisms. We also decided to include only those admitted to the center and not others initially admitted in another healthcare facility, which might explain some of the lower scores. However, this decision may have also helped diminish confounding factors due to complex and unknown treatment courses outside the studied facility, cross-contamination, and nosocomial infections.

## 4. Materials and Methods

### 4.1. Study Design and Setting

A retrospective cohort study was performed in Clínica Reina Sofia (Clínicas Colsanitas), a third-level-complexity hospital in Bogotá, Colombia. It serves a population of around 500,000 individuals from Bogotá, Colombia, and other cities. We included information obtained from adult patients hospitalized with a positive urinary tract culture for *E. coli* collected in the emergency department who had been clinically diagnosed with pyelonephritis between 1 January 2013 and 30 June 2020. *E. coli* isolates were chosen because they corresponded to 88% of the Enterobacterales-producing urinary tract infections in a previous study in Colombia and patients with *K. pneumoniae* can more frequently have an antecedent of nosocomial infection or harbor a different resistant mechanism (such as carbapenemase production) [31]. Patients were excluded if they had incomplete clinical records, the antibiotic was changed in the first 24 h, they used a permanent urinary catheter, susceptibility testing was not performed, or no data on outcomes were available because of transferal to a different hospital.

#### 4.1.1. Variables and Definitions

##### Microbiology

Information was obtained through the WHONET platform (WHO ver 5.6). Clinical samples were processed in Colsanitas’ central laboratory, using automated VITEK 2 (bioMérieux, Marcy-l’Étoile, France) microbiological growth systems. ESBL detection was performed according to the Clinical Laboratory Standards Institute (CLSI, Malvern, PA, USA), and no changes to the interpretation of ESBL production were made over the course of the study. Non-susceptible isolates refer to the combination of the categories intermediate and resistant to ceftriaxone (third-generation cephalosporin) or cefuroxime (second-generation cephalosporin).

##### Variables

Information was obtained from the electronic medical records of the healthcare facility. The variables included were based on Tumbarello’s original article [10] and associated publications. The following types of data were analyzed: 1. patient demographic data and 2. information from electronic medical records obtained at admission, including data on immunosuppression, antibiotic and/or urinary catheter usage in the previous 3 months, urinary interventions in the previous 3 months, and hospitalization in the previous year. Data were collected to calculate the Charlson score to evaluate comorbidity and a potential relationship to resistance. Patients with admission from another healthcare facility were not included because such infections more commonly represent a nosocomial infection and the aim was to evaluate the impact of the infections acquired in the community.

##### Statistical Analysis

Categorical variables were analyzed using χ^2^ tests and continuous variables were analyzed according to their distribution, using Student’s *t*-test or the Wilcoxon test accordingly. Prediction rules were applied to our patient population and compared to the microbiological results. The primary outcomes were the sensitivity (true positives over ESBL producers), specificity (true negatives over non ESBL producers), positive predictive value (PPV, true positives over all positive results), and negative predictive value (NPV, true negatives over all negative results) of the clinical prediction rules for an ESBL-producing microorganism. A receiver operating characteristic (ROC) curve was generated for the index, and the area under the ROC (AUC ROC) curve was calculated to determine discriminative power. The Stata (version 15.1, Statacorp, College Station, TX, USA) and R software (version 4.0.2, R Foundation, Vienna, Austria) were used for analyses. Other sensitivity analyses were performed using any resistance to third-generation cephalosporins and non-sensitivity to second-generation cephalosporins.

## 5. Conclusions

In conclusion, our study shows that in a high-prevalence area such as Colombia, the score proposed by Tumbarello et al. [10] has low sensitivity and accuracy, possibly favoring the use of broad-spectrum antibiotics like carbapenems in scenarios where they can be avoided. This potentially contributes to the inappropriate use of this group of beta-lactams and further increases in different forms of antimicrobial resistance.

## Figures and Tables

**Figure 1 antibiotics-14-00938-f001:**
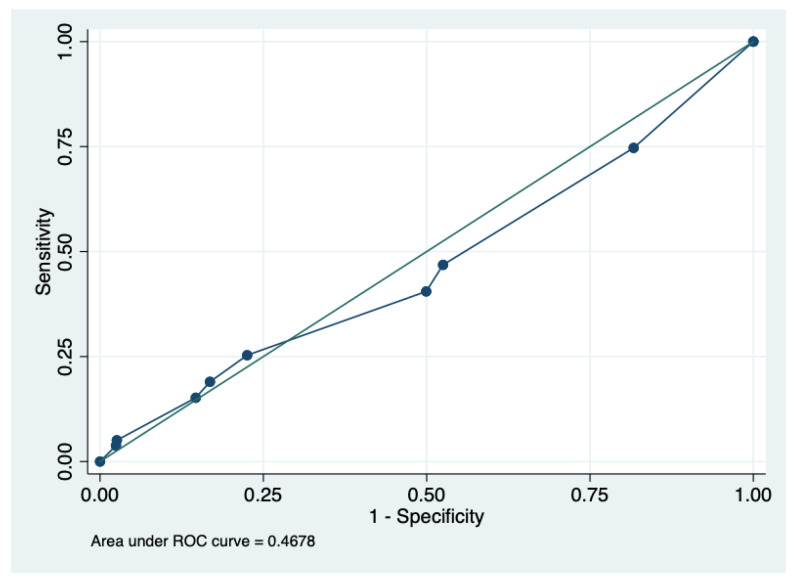
The receiver operating characteristic area under the curve for predicting ESBL phenotype using the score. ESBL: extended-spectrum beta-lactamase.

**Table 1 antibiotics-14-00938-t001:** Baseline characteristics of the patients and risk factors.

Characteristics	Non ESBL	ESBL	Total	*p* Value
**Demographical**				
Male, *n* (%)	227 (30.8)	42 (53.2)	270 (33.0)	<0.001
Age, mean (SD)	64.0 (19.4)	65.6 (15.6)	64.2 (19.1)	0.492
Older than 70 years, *n* (%)	337 (45.7)	31 (39.2)	369 (45.2)	0.326
**Past medical information**				
Diabetes, *n* (%)	136 (18.5)	14 (17.7)	151 (18.5)	0.995
Charlson, mean (SD)	4.0 (1.8)	3.3 (1.8)	3.9 (1.8)	0.001
Previous antibiotic use, *n* (%)	139 (18.9)	18 (22.8)	157 (19.2)	0.490
Previous beta-lactam use, *n* (%)	70 (9.5)	11 (13.9)	81 (9.9)	0.293
Previous quinolone use, *n* (%)	24 (3.3)	5 (6.3)	29 (3.5)	0.279
Previously hospitalized, *n* (%)	168 (22.8)	21 (26.6)	189 (23.1)	0.537
Previous urinary catheter, *n* (%)	19 (2.6)	7 (8.9)	26 (3.2)	0.007

SD: Standard deviation; ESBL: extended-spectrum beta-lactamase.

**Table 2 antibiotics-14-00938-t002:** Diagnostic performance of the Tumbarello score in assessing *E. coli* isolates from community-acquired urinary tract infections.

**Prediction of ESBL Phenotype**									
**Score≥**	**TP**	**FP**	**TN**	**FN**	**Sens (%)**	**Esp (%)**	**PPV (%)**	**NPV (%)**	**ACC (%)**
2	59	602	135	20	74.7	18.3	8.9	87.1	23.8
3	37	387	350	42	46.8	47.4	8.7	89.2	47.4
4	32	368	369	47	40.5	50	8.0	88.7	49.1
5	20	166	571	59	25.3	77.4	10.7	90.6	72.4
6	15	124	613	64	18.9	83.1	10.7	90.5	76.9
7	12	108	629	67	15.1	85.3	10.0	90.3	78.5
8	4	19	718	75	5.1	97.4	17.3	90.5	88.4
9	3	18	719	76	3.7	97.5	14.2	90.4	88.4
**Prediction of Third-Generation Cephalosporin Non-Susceptibility**									
**Score≥**	**TP**	**FP**	**TN**	**FN**	**Sens (%)**	**Esp (%)**	**PPV (%)**	**NPV (%)**	**ACC (%)**
2	83	579	128	27	75.4	18.1	12.5	82.5	25.8
3	56	369	338	54	50.9	47.8	13.1	86.2	48.2
4	51	350	357	59	46.3	50.4	12.7	85.8	49.9
5	27	159	548	83	24.5	77.5	14.5	86.8	70.3
6	22	117	590	88	20.0	83.4	15.8	87.0	74.9
7	19	101	606	91	17.2	85.7	15.8	86.9	76.4
8	6	17	690	104	5.4	97.5	26.0	86.9	85.1
9	5	16	691	105	4.5	97.7	23.8	86.8	85.1

The upper section indicates performance based on ESBL reporting, while the lower section indicates performance in predicting non-susceptibility to third-generation cephalosporins. Abbreviations: TP, true positive; FP, false positive; TN, true negative; FN, false negative. Given in percentages: Sens, sensibility; Esp, specificity. PPV, positive predictive value; NPV, negative predictive value; ACC%, accuracy. The cut-off (score of 6 points) was selected according to the original publication by Tumbarello et al. [10].

## Data Availability

Anonymized data are available on request.

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
