# Peer review of "Prediction Score for Identification of ESBL Producers in Urinary Infections Overestimates Risk in High-ESBL-Prevalence Setting"

_antibiotics, 2025, doi:10.3390/antibiotics14090938_

Round 1
Reviewer 1 Report
Comments and Suggestions for Authors
This retrospective cohort study addresses the validation of an established ESBL prediction score in Colombia's high-prevalence setting for Escherichia coli UTIs, highlighting the drawbacks of using instruments designed for low-prevalence environments in high-prevalence areas. The manuscript is ethically acceptable, methodologically sound, and clear.
- The study highlights the ineffectiveness of an existing ESBL risk tool in Colombia and provides valuable regional data for stewardship initiatives. However, it lacks detailed discussion on alternative tactics or adjustments. A discussion paragraph is recommended to explore future strategies based on the study's findings.
- The authors changed the Tumbarello score by eliminating a variable, admission from another healthcare facility, without providing a justification. It is recommended to clearly explain the methodological change and discuss potential effects.
- The paper warns against overuse of broad-spectrum antibiotics but does not provide recommendations for alternatives. It suggests clinicians use biomarkers, local antibiograms, or decision support tools in high-prevalence situations.
Minor comments
- Numerous instances of improper grammar or word usage (for example, "microoorganisms" →"microorganisms"; "hidrolyse" should be "hydrolyze").
→ Recommend a professional service or a native speaker to edit all of the English text.
Terminology: "Non-susceptible" and "resistant" should be defined precisely and used consistently.
Presentation of Figures and Tables
The provided text mentions but does not display Figure 1 (ROC curve). Make sure every table and figure has the correct label and is included.
Tables provide useful information, but legends—especially Table 2—need to be clarified. Describe the selection process for cutoffs and abbreviations.
Author Response
Quality of English Language: The English could be improved to more clearly express the research.
Answer: A professional English editing service was used to improve the language.
This retrospective cohort study addresses the validation of an established ESBL prediction score in Colombia's high-prevalence setting for Escherichia coli UTIs, highlighting the drawbacks of using instruments designed for low-prevalence environments in high-prevalence areas. The manuscript is ethically acceptable, methodologically sound, and clear.
- The study highlights the ineffectiveness of an existing ESBL risk tool in Colombia and provides valuable regional data for stewardship initiatives. However, it lacks detailed discussion on alternative tactics or adjustments. A discussion paragraph is recommended to explore future strategies based on the study's findings.
Answer: Thank you for the observation. A complete paragraph of the current findings and prospective was added (Lines 145-163).
- The authors changed the Tumbarello score by eliminating a variable, admission from another healthcare facility, without providing a justification. It is recommended to clearly explain the methodological change and discuss potential effects.
Answer: Thank you for the observation. The admission from other healthcare facility was not included due to the fact that these group of patients would reflect nosocomial infections instead of those acquired in the community. This was added in the methodology. We also acknowledge this fact in the limitations.
- The paper warns against overuse of broad-spectrum antibiotics but does not provide recommendations for alternatives. It suggests clinicians use biomarkers, local antibiograms, or decision support tools in high-prevalence situations.
Answer: We suggest in the meantime to adhere to guidelines that promote the use of antibiotics with higher concentration in the urine, taking into account that nonsusceptibility in this scenario was not related to clinical failure. We added some lines (139-144) to the discussion to emphasize this point in the discussion.
Minor comments
- Numerous instances of improper grammar or word usage (for example, "microoorganisms" →"microorganisms"; "hidrolyse" should be "hydrolyze").
→ Recommend a professional service or a native speaker to edit all of the English text.
Terminology: "Non-susceptible" and "resistant" should be defined precisely and used consistently.
Answer: A professional English editing service was used to improve the language and solve typos and inconsistent phrasing. The terminology was adjusted and used consistently. The definition was added in the methodology.
- Presentation of Figures and Tables
The provided text mentions but does not display Figure 1 (ROC curve). Make sure every table and figure has the correct label and is included.
Answer: We included the figure and the checked for labels.
- Tables provide useful information, but legends—especially Table 2—need to be clarified. Describe the selection process for cutoffs and abbreviations.
Answer: The reviewed the label for table 2 and tired to simplify the information.The abbreviations are described.
Reviewer 2 Report
Comments and Suggestions for Authors
ESBL-producing E. coli and third-generation cephalosporin resistance are highly relevant in antimicrobial management, especially in high-prevalence regions like Colombia. Evaluating the Tumbarello score’s diagnostic performance in community UTIs is a valuable contribution, especially if it informs antibiotic decision-making. But the manuscript lacks clarity and there are some major issues to address.
Major issues:
- The manuscript is difficult to understand and needs proofreading (e.g. There are multiple typographical errors (e.g., line “loke” instead of “like”, ”.”in place of “,” in table 2) and clarity and flow throughout the text.
- study does not include strong statistical methods. It lacks multivariable analysis to show which factors are truly linked to ESBL infection. Many p-values (e.g., for diabetes, age, prior antibiotic use) are not significant, but the discussion does not explain this or explore deeper analysis.
- Tumbarello score shows poor sensitivity across cut-offs (e.g., 18.9% at score ≥6), which undermines its utility in early detection, but the manuscript should explore alternative scoring systems through analysis of existing data or to acquire data to improve sensitivity.
Author Response
Quality of English Language: The English could be improved to more clearly express the research.
Answer: A professional English editing service was used to improve the language.
ESBL-producing E. coli and third-generation cephalosporin resistance are highly relevant in antimicrobial management, especially in high-prevalence regions like Colombia. Evaluating the Tumbarello score’s diagnostic performance in community UTIs is a valuable contribution, especially if it informs antibiotic decision-making. But the manuscript lacks clarity and there are some major issues to address.
Major issues:
- The manuscript is difficult to understand and needs proofreading (e.g. There are multiple typographical errors (e.g., line “loke” instead of “like”, ”.”in place of “,” in table 2) and clarity and flow throughout the text.
Answer: A professional English editing service was used to improve the language.
- study does not include strong statistical methods. It lacks multivariable analysis to show which factors are truly linked to ESBL infection. Many p-values (e.g., for diabetes, age, prior antibiotic use) are not significant, but the discussion does not explain this or explore deeper analysis.
Answer: A multivariable analysis was not performed, since the aim of the study was not to create a different or new score, but to evaluate the prediction capacity of the score proposed by Tumbarello et al, since it is commonly used in the country. A paragraph on the differences found in the bivariate analysis was added to the discussion. Lines 102-108.
- Tumbarello score shows poor sensitivity across cut-offs (e.g., 18.9% at score ≥6), which undermines its utility in early detection, but the manuscript should explore alternative scoring systems through analysis of existing data or to acquire data to improve sensitivity.
Answer: Thank you for the observation. We added a prapgraph on the current options that are being explored by researchers.Lines 145-163
Reviewer 3 Report
Comments and Suggestions for Authors
This retrospective single center study from a Colombian tertiary hospital assesses the performance of the Tumbarello clinical prediction score originally developed for low to moderate prevalence settings in predicting ESBL producing E. coli in urinary tract infections (UTIs) in a high ESBL prevalence context. The dataset spans 2014-2021 and includes 1,029 E. coli urinary isolates from first-admission episodes. The study compares both the original and a locally modified version of the score (with “admission from another healthcare facility” removed), reporting diagnostic performance metrics (ROC-AUC, sensitivity, specificity, predictive values) for ESBL production and non-susceptibility to multiple antibiotics.
This article demonstrates scientific soundness and is eligible for publication with a few suggested improvements:
Methodological Clarifications
-
Line 104: Clarify rationale for limiting to E. coli, exclude other Enterobacterales for methodological validity or epidemiology?
-
Line 127: Provide fuller justification for removing “admission from another healthcare facility” from the score. Was it absent from records, or found to have poor predictive value?
Interpretation of Score Performance
-
Line 211: Sensitivity at ≥6 cut-off is only 18.9% too low for clinical decision making. Discuss whether lowering the threshold or adopting a dual cut-off strategy might improve utility.
Stewardship Implications
-
Lines 280–295: Expand discussion on clinical implications, could carbapenem sparing regimens be considered for moderate-score patients in high prevalence contexts? Address the risk of undertreatment if scores are disregarded entirely.
Statistical Context
-
Include a note on whether sample size was adequate for detecting meaningful ROC-AUC differences (retrospective power analysis if possible).
Minor Editorial Refinements:
-
Line 41: “loke” → “like”
-
Line 62: “hidrolyse” → “hydrolyze” (or “hydrolyse” if using UK spelling, please ensure consistency).
-
Table 1: Clarify if “Other antibiotic use” refers to systemic antibiotics in the preceding 30 days.
-
Table 2: Add 95% CI for sensitivity, specificity, PPV, NPV.
-
Figures: Define all abbreviations in legends (e.g. ESBL, NS).
-
Check rounding consistency between text and tables.
This is a strong and relevant validation study for a high prevalence AMR setting. The dataset is robust, the methodology sound, and the conclusions well supported. With modest revisions, focused on methodological clarity, stewardship implications, and minor editorial fixes, the manuscript will make a valuable contribution to the scientific community.
Author Response
Methodological Clarifications
- Line 104: Clarify rationale for limiting to coli, exclude other Enterobacterales for methodological validity or epidemiology?
Answer: E. coli isolates were chosen because of they correspond to 88% of the Enterobacterales producing urinary tract infection in a previous study in Colombia and K. pneumoniae more frequently can have an antecedent of nosocomial infection or harvour a different resistant mechanism (such as carbapenemase production) that might bias the result. This was added to the methods section (lines 183-187).
- Line 127: Provide fuller justification for removing “admission from another healthcare facility” from the score. Was it absent from records, or found to have poor predictive value?
Answer: Thank you for the observation. The admission from other healthcare facility was not included due to the fact that these group of patients would reflect nosocomial infections instead of those acquired in the community. This was added in the methodology. We also acknowledge this fact in the limitations.
Interpretation of Score Performance
- Line 211: Sensitivity at ≥6 cut-off is only 18.9% too low for clinical decision making. Discuss whether lowering the threshold or adopting a dual cut-off strategy might improve utility.
Answer: The reviewer is right. We added some interpretation of the results and the ROC curve to the discussion (Lines 125-128 ).
Stewardship Implications
- Lines 280–295: Expand discussion on clinical implications, could carbapenem sparing regimens be considered for moderate-score patients in high prevalence contexts? Address the risk of undertreatment if scores are disregarded entirely.
Answer: Thank you for the observation. We added some lines to the paragraph in the discussion with alternatives to the use of predicting scores (lines 139-144).
Statistical Context
- Include a note on whether sample size was adequate for detecting meaningful ROC-AUC differences (retrospective power analysis if possible).
Answer: 95% CI for ROC AUC were added in the text. The sample power was added. Since the ROC AUC was close to 0.5 the power is limited.
Minor Editorial Refinements:
- Line 41: “loke” → “like”
- Line 62: “hidrolyse” → “hydrolyze” (or “hydrolyse” if using UK spelling, please ensure consistency).
- Table 1: Clarify if “Other antibiotic use” refers to systemic antibiotics in the preceding 30 days.
- Table 2: Add 95% CI for sensitivity, specificity, PPV, NPV.
- Figures: Define all abbreviations in legends (e.g. ESBL, NS).
- Check rounding consistency between text and tables.
The text was reviewed with professional editing service to correct all the typos and misspelling. 95% CI were not added since the table would be too crowded. Legends with abbreviations were added to tables and figures. The rounding was checked for consistency in the tables.
Round 2
Reviewer 2 Report
Comments and Suggestions for Authors
The authors have improved the manuscript readability, making it acceptable for publication. The manuscript also highlights the importance of the current methods’ limitations or failures. However, a few questions arose while reading it.
- Why was the study conducted in a single hospital in Bogotá, Colombia? How does this affect the generalizability of the results?
- What further studies are needed to develop accurate tools for assessing the risk of ESBL producers in high-prevalence settings? Authors could add in discussion section
- Line 11, “produced by Extended-Spectrum Beta-lactamase". That doesn't seem quite right. UTIs aren't "produced by" ESBL, they're caused by bacteria that produce ESBL. Maybe "caused by" would be better?
- "With an index cutoff of >=6, the score showed a sensitivity of..."; symbol ">=" is more commonly used in programming. In formal writing, it's better to write out "greater than or equal to" or use the proper mathematical symbol (≥).
- line 208, The introduction of “the Charlson score” is stated but not explained. Without detailing how the Charlson score is used, it's difficult to assess the study’s relevance to broader clinical context and potential impact on patient management
- line 216, The text mentions “sensitivity, specificity, positive predictive value, and negative predictive value” for the clinical prediction rules, but it doesn't explain how these are calculated or what they measure. The absence of this explanation raises concerns about the interpretability of the results
- Line 179, “a private insurance company (Colsanitas- prepagada)” is redundant and could be removed.
Author Response
Please find attached responses.
